# Evaluation of the 50% Infectious Dose of Human Norovirus Cin-2 in Gnotobiotic Pigs: A Comparison of Classical and Contemporary Methods for Endpoint Estimation

**DOI:** 10.3390/v12090955

**Published:** 2020-08-28

**Authors:** Ashwin K. Ramesh, Viviana Parreño, Philip J. Schmidt, Shaohua Lei, Weiming Zhong, Xi Jiang, Monica B. Emelko, Lijuan Yuan

**Affiliations:** 1Department of Biomedical Sciences and Pathobiology, Virginia-Maryland College of Veterinary Medicine, Virginia Tech, Blacksburg, VA 24060, USA; akramesh@vt.edu (A.K.R.); lsh2013@vt.edu (S.L.); 2Instituto de Virología, Instituto Nacional de Tecnología Agropecuaria (INTA), Castelar 1712, Argentina; parreno.viviana@inta.gob.ar; 3Water Science, Technology & Policy Group, Department of Civil & Environmental Engineering, University of Waterloo, Waterloo, ON N2L 3G1, Canada; pj2schmidt@uwaterloo.ca (P.J.S.); mbemelko@uwaterloo.ca (M.B.E.); 4Division of Infectious Diseases, Cincinnati Children’s Hospital Medical Center, Cincinnati, OH 45229, USA; Weiming.Zhong@cchmc.org (W.Z.); Jason.Jiang@cchmc.org (X.J.); 5Department of Pediatrics, University of Cincinnati College of Medicine, Cincinnati, OH 45229, USA

**Keywords:** median infectious dose, Reed–Muench, Spearman–Karber, approximate beta-Poisson, dose–response

## Abstract

Human noroviruses (HuNoVs) are the leading causative agents of epidemic and sporadic acute gastroenteritis that affect people of all ages worldwide. However, very few dose–response studies have been carried out to determine the median infectious dose of HuNoVs. In this study, we evaluated the median infectious dose (ID_50_) and diarrhea dose (DD_50_) of the GII.4/2003 variant of HuNoV (Cin-2) in the gnotobiotic pig model of HuNoV infection and disease. Using various mathematical approaches (Reed–Muench, Dragstedt–Behrens, Spearman–Karber, logistic regression, and exponential and approximate beta-Poisson dose–response models), we estimated the ID_50_ and DD_50_ to be between 2400–3400 RNA copies, and 21,000–38,000 RNA copies, respectively. Contemporary dose–response models offer greater flexibility and accuracy in estimating ID_50_. In contrast to classical methods of endpoint estimation, dose–response modelling allows seamless analyses of data that may include inconsistent dilution factors between doses or numbers of subjects per dose group, or small numbers of subjects. Although this investigation is consistent with state-of-the-art ID_50_ determinations and offers an advancement in clinical data analysis, it is important to underscore that such analyses remain confounded by pathogen aggregation. Regardless, challenging virus strain ID_50_ determination is crucial for identifying the true infectiousness of HuNoVs and for the accurate evaluation of protective efficacies in pre-clinical studies of therapeutics, vaccines and other prophylactics using this reliable animal model.

## 1. Introduction

Human noroviruses (HuNoVs) are non-enveloped RNA viruses with a positive-sense single-stranded genome that belong in the *Caliciviridae* family. They are the most common etiological agents of epidemic and sporadic acute gastroenteritis in people of all age groups [1,2,3] and are responsible for around 125 million cases of foodborne illnesses worldwide each year [4]. There are currently no licensed vaccines available for HuNoVs, but several virus-like particle (VLP) vaccine candidates are currently undergoing various stages of clinical trials and have thus far shown some promise in both immunogenicity, and protective efficacy studies, although the correlates of protection to norovirus infections are yet to be fully determined [5,6,7,8]. Studies of determinants of protective immunity in humans is difficult and costly, and preclinical studies of vaccines and therapeutics in animal models are an essential step before regulatory approval of human clinical trials.

An ideal animal model of HuNoV infection and disease would be one that can replicate the biological and clinical features associated with human disease, including the route of infection, the infective dose, the disease progression and pathogenesis, and correlates of protection [9]. Among the different models used in norovirus (NoV) research [10,11,12,13,14], gnotobiotic (Gn) pigs provide a suitable solution for studying HuNoV infection and disease, as well as for evaluating immunogenicity and protective the efficacy of novel vaccine candidates. This is attributed to the close similarity in physiology, immune development, virus binding patterns, and histo-blood group antigen (HBGA) phenotypes between pigs and humans [11]. Since their introduction into research in the early 1960s, Gn pigs have been extensively utilized for the study of enteric viruses and bacteria [15,16]. A Gn pig infection model for two different HuNoV GII.4 variants (GII.4/HS66 and GII.4/2006b) has been established for evaluating HuNoV vaccines [17,18,19,20], displaying similar levels of infectivity compared to HuNoV GI.1/Norwalk and GII.4/2003 Cin-1 in human challenge studies and vaccine trials [7,8,21].

HuNoVs are known to be highly infectious. A few challenge studies have evaluated the infection potential [22,23,24] and calculated the median infectious dose (ID_50_) of various norovirus strains. In humans, the ID_50_ of Norwalk virus was identified to be between 18 and 2800 genomic equivalents [25,26]. Similarly, the ID_50_ of a GII.4/2006b variant was identified to be ≤2.74 × 10^3^ RNA copies in newborns (4–5 days of age) and 6.43 × 10^4^ RNA copies in older (33–34 days of age) Gn pigs [18]. Dose–response data are critically important for the standardization of the animal challenge model used in the pre-clinical evaluation of vaccine efficacy and anti-viral agents. While ID_50_ is of primary interest to clinicians, such studies are also widely used to interpret pathogen occurrence and exposure data and translate them to health outcomes. For example, quantitative microbial risk assessments (QMRAs) for pathogen infections in humans are regularly used to develop and improve industrial and regulatory policy in the water and food safety sectors [27,28]. Utilization of dose–response approaches (experimental design and analysis) concurrently enables wider application and impact from experimentally derived data.

ID_50_ values are often estimated using classical methods, such as the Reed–Muench, Dragstedt–Behrens, and Spearman–Karber methods or more recently by logistic regression. In recent decades, several mechanistic models have been developed to describe plausible phenomena that are inherent to experimentally derived dose–response data [25,29]. Critically, such models provide a basis for inference about the probability of infection at any dose level (i.e., not just ID_50_), though low-dose extrapolation is an ever-present concern for applications, such as drinking water risk assessments in which mean doses are often less than one pathogen [30]. Contemporary dose–response models may offer greater flexibility and accuracy (i.e., managing inconsistent dilution factors between doses or numbers of subjects per dose group or small numbers of subjects) in estimating ID_50_.

The objectives of this study were to determine the median infectious and diarrhea doses—the dilutions of virus inoculum at which 50% of the Gn pigs become infected (ID_50_) or develop diarrhea (DD_50_)—when Gn pigs are infected with a pandemic strain of HuNoV. The significance of this study includes: (i) experimental evaluation of GII.4/2003 HuNoV dose–response in 33–34 day old Gn pigs; (ii) comparison of different dose–response analyses used for estimating the ID_50_ and DD_50_ of HuNoV in Gn pigs to determine the best-fit model; (iii) identification of the most appropriate challenge dose in 33–34 day old Gn pigs to standardize the model for HuNoV vaccine evaluation; (iv) comparison of the obtained ID_50_ and the ID_50_ used in human volunteer challenge studies. Comparing the infectiousness of the GII.4 variant challenge pools in Gn pigs and humans will lead to a better understanding of the zoonotic potential [31] of NoVs between different species and further validate the Gn pig model of HuNoV infection as a proper predictive tool of the future efficacy of vaccines and other antiviral strategies to control NoV diarrhea in humans.

## 2. Materials and Methods

### 2.1. Virus Inoculum

The GII.4 HuNoV challenge pool (103,041) used in this study, identified henceforth as Cin-2, was the unfiltered 10% suspension of stool samples collected from a volunteer who was challenged with HuNoV GII.4/2003 (Hu/GII.4/Cin-1/2003/US), a 2002 Farmington Hills-like variant (GenBank number JQ965810), as part of a vaccine study conducted by Xi Jiang’s laboratory at Cincinnati Children’s Hospital Medical Center. The volunteer developed a 3-day illness characterized by diarrhea, vomiting, nausea, abdominal cramps, and fever [7,32]. The HuNoV concentration of the challenge pool was determined by RT-qPCR to be approximately 2 × 10^6^ viral genomic RNA copies/mL of stool. The challenge pool was stored at −80 °C in individual 1 mL aliquots until the day of Gn pig inoculation. The Gn pig challenge studies were conducted between July 2017 and March 2018.

### 2.2. Gnotobiotic Pigs and Treatments

Near-term pigs (large white cross breed) were derived by aseptic hysterectomy and maintained in sterile isolator units as described previously [15]. To determine the sterility status of the isolators in which the pigs were housed, rectal swabs collected from pigs were cultured on blood agar plates (Hardy Diagnostics, Santa Maria, CA, USA) and in Thioglycollate media (Hardy Diagnostics, Santa Maria, CA, USA) 3 days after derivation and repeated once a week until the end of the study. Pigs were fed commercial ultra-high temperature-treated sterile cow milk throughout the study. A total of 28 pigs were assigned to seven groups. Pigs were randomly divided into groups upon derivation regardless of gender and body weight. Each group of pigs received a different dose of Cin-2 inoculum at postpartum day (PPD) 33 or 34. Four milliliters of 200 mM sodium bicarbonate was given orally 15 min prior to inoculation with HuNoV to neutralize stomach acids. Doses were prepared by serial dilution, in PBS, of the initial virus stock into subsequent doses and vortexed to ensure thorough mixing. The inoculum was prepared and added to 5 mL of Diluent #5 (minimal essential medium with 1% penicillin-streptomycin and 1% HEPES) on the day of inoculation. All pigs received oral doses of Cin-2 ranging from 8 × 10^2^ to 2 × 10^6^ genomic RNA copies at PID (post-inoculation day) 0. These doses were determined based on doses used in human volunteer studies [7,26,32] as well as work carried out by our group using Gn pigs [18]. Diarrhea and fecal virus shedding were monitored daily until euthanasia. All pigs were euthanized on PID 7. All experiments involving the use of Gn pigs were approved by the Institutional Animal Care and Use Committee at Virginia Tech (IACUC protocol: 17-110-CVM). All experimental procedures were carried out in compliance with federal and university regulations.

### 2.3. HBGA-Typing of Gn Pigs by Immunofluorescence Assay

Prior to inoculation, all Gn pigs were screened for their HBGA type by immunofluorescence assay. Epithelial cells collected from the cheeks were prepared as described previously [18] and HBGA phenotypes were detected from these cells using fluorescent labeled antibodies described elsewhere [33]. All slides were prepared with VECTASHIELD^®^ Antifade Mounting Medium with 4,6-diamidino-2-phenylindole (DAPI), which was used as a nuclear counterstain (Vector Laboratories, Burlingame, CA, USA) for fluorescent microscopy. Based on this screening technique, HBGA type A^−^ and H^−^ pigs were excluded from this study because of their reduced susceptibility to NoV infection relative to A^+^ or H^+^ pigs [11,18].

### 2.4. Assessment of Fecal Consistency and Detection of HuNoV Shedding by RT-qPCR

Fecal consistency and virus shedding were monitored at 24-h intervals at the same time daily, after HuNoV inoculation at PID 0, by rectal swab sampling. Fecal consistency scores were determined based on previous studies carried out by our group [33,34,35]: 0 solid; 1 pasty; 2 semi-liquid; 3 liquid. Pigs with a fecal consistency score of 2 or greater (≥2) were considered to be diarrheic. Presence of HuNoV genomes in rectal swab samples collected after inoculation with different doses of Cin-2 were evaluated using RT-qPCR protocol as described previously [33]. Briefly, once collected, swabs were swirled in 1 mL PBS to release the feces and centrifuged at 10,000× *g* for 5 min. Then, 250 µL of the supernatant collected was mixed with 750 μL TRIzol (Invitrogen, Carlsbad, CA, USA) following the manufacturer’s instructions for viral RNA extraction. The extracted RNA was then dissolved in 40 μL autoclaved dd-H_2_O. Five microliters of RNA was used in a 20 μL RT-qPCR reaction with a SensiFAST Probe No-ROX One-Step Kit (Bioline, London, UK) to detect HuNoV genomes. Primers COG2F and COG2R, and probe RING2, were used with cycling conditions as described in a previous study [36]. A standard curve was generated using COG2 amplicon-containing cDNA expression plasmids in standards serially diluted tenfold from 2 × 10^6^ to 2 genomic RNA copies. Amplification was performed on CFX96 Real-Time System (Bio-Rad, Hercules, CA, USA), and data were collected and analyzed with Bio-Rad CFX Manager 2.0.

### 2.5. Statistical Analysis

#### 2.5.1. Analysis of Variance among Challenge Doses

The statistical analyses of variances were carried out using Infostat^®^ (https://www.infostat.com.ar) connected to R software (R Core Team) in R Studio (see Appendix A). The variables related to virus shedding were analyzed using a generalized linear mixed model (GLMM), considering the treatment (pigs receiving different doses of HuNoV) as a fixed variable and the Gn pigs within the groups as a random variable. A varIdent structure for the variance–covariance matrix was used for modeling the heterogeneity of variance. The variables associated to diarrhea that met normality and homoscedasticity assumptions were analyzed by one-way ANOVA. In all cases, multiple post-ANOVA comparisons were carried out using the Tukey method for comparisons between the 7 different doses. Statistical significance was considered at *p* < 0.05 for all comparisons (Appendix A and Table 1).

#### 2.5.2. ID_50_ and DD_50_ Calculations Using Different Approaches

For a specific pathogen, host, exposure route, and endpoint (e.g., virus shedding or diarrhea in this work), there is a potentially unique dose–response relationship. Classical methods for analyzing dose–response data provide rudimentary estimation of ID_50_—they were developed to facilitate ease of analysis before widespread availability of computing. Statistical methods, such as logistic regression, allow the estimation of dose–response at any dose and seamless analysis of data with doses that are not equally spaced (in logarithmic scale) or with variable numbers of subjects per group, but ultimately provide only an empirical fit. Exponential and exact beta-Poisson dose–response models share the advantages of logistic regression but also account for known mechanisms such as Poisson-distributed numbers of pathogens in a sample of known volume and concentration if the pathogens are disaggregated.

##### Reed–Muench, Dragstedt–Behrens and Spearman–Karber Methods

The Reed–Muench (RM) method, first described in 1938 [37], is an old fashioned method that has been widely used for the calculation of 50% end points in experimental biology. The RM method employs a two-step equation that involves the calculation of the proportionate distance (PD) between dilutions above and below the 50% end point, as calculated using Equation (1).
(1)$$\left(\mathrm{PD}\right)=\;\frac{\mathrm{Percentage}\;\mathrm{infected}\;\mathrm{at}\;\mathrm{dilution}\;\mathrm{next}\;\mathrm{above}\;50\%-50\%}{\mathrm{Percentage}\;\mathrm{infected}\;\mathrm{at}\;\mathrm{dilution}\;\mathrm{next}\;\mathrm{above}\;50\%-\mathrm{Percentage}\;\mathrm{infected}\;\mathrm{at}\;\mathrm{dilution}\;\mathrm{next}\;\mathrm{below}\;50\%}$$
(2)log50% end point=(log dilution above 50%)−(PD∗log dilution factor)

The dilution factor used in this equation describes the fold difference between the two inoculum titers above and below a 50% response rate. The median dose was then calculated based on Equation (2). The RM method were used previously for the determination of the ID_50_ of a different HuNoV variant (GII.4/2006b) in Gn pigs [18].

The Dragstedt–Behrens (DB) method [38,39] is very similar to the RM method but instead of working with the cumulative sums directly, it uses the frequency of the cumulative sums that are positive at each dose. The ID_50_ is estimated by interpolating the line that connects the hypothetical fractions of the bracketing doses [40].

Both methods rely on doses to be equally-spaced logarithmically, with each group containing equal numbers of subjects (e.g., pigs) for an accurate estimate of the dose–response. Although primarily used for their historical relevance, the RM and DB methods are still used extensively in modern scientific research as the primary approach for calculating the end-point dilution dose [18,41,42,43,44,45,46].

Spearman–Karber is another method that is widely used and specially suggested by regulatory agencies for the calculation of virus titers and median infectious doses [47]. An important criterion for being able to use this method is the presence of dose–response data with 0% and 100% responses in order to generate a symmetrical distribution of data with well-defined upper and lower plateaus [40]. All these methods were calculated by hand and using the skrmdb package in R [47].

##### Exponential and Beta-Poisson Dose–Response Models

Exponential and beta-Poisson models are widely recognized as suitable models for describing dose–response for various viral infections in humans and other animal models. They include the assumption that one virus particle is sufficient to cause an infection if it arrives at an appropriate site. They also include the assumption that the viruses are disaggregated with homogeneous concentration, such that the exact number of viruses administered in a dose is Poisson-distributed. Meeting these assumptions enables for the mechanistic description of the effects of dose and host susceptibility upon dose–response. When they are not met (e.g., when viruses are aggregated), these models may still be useful to fit/describe the empirical relationship, but this precludes mechanistic inference. The exponential dose–response model in Equation (3) is derived by assuming that the parameter r representing the probability of survival of the pathogen in the host is a constant among hosts. Specifically, the number of ingested pathogens surviving to initiate infection is binomially distributed with survival probability r.
(3)P(r)=1−exp(−r.dose)

The exact beta-Poisson dose–response model is an expansion from the exponential model in which it is assumed that the probability of survival of each pathogen in the host (r) varies randomly among hosts or exposures, according to a beta distribution. This allows for an increased level of realism in modeling the dose–response relationship in the host. Equation (4) represents a common approximation of the exact form of the beta-Poisson model [27], where β = N_50_/[$2^{\frac{1}{\alpha }}-1$]. The criteria β >> α and β >> 1 are well satisfied with the model fits obtained from this study.
(4)$$\left(r\right)=1-{\left[1+\frac{dose\cdot \left(2^{\frac{1}{\alpha }}-1\right)}{N_{50}}\right]}^{-\alpha }$$

In this equation, *N*_50_ is an estimate of the median infectious/diarrhea dose with units of counts (i.e., genome copies) and *α* is a unitless model parameter. This approximated form retains linearity at low doses, a fundamental characteristic of these microbial dose–response models [27,48].

An online tool is available for the– exponential dose response model, logistic regression, and the Spearman–Karber method, and can be found here: NCBI Statistical Computational Biology Infectious Dose or Dilution (ID_50_) Server—https://www.ncbi.nlm.nih.gov/CBBresearch/Spouge/html_ncbi/html/id50/id50.cgi [49]. This tool was based on VACMAN, a computational program that calculates statistics for in vitro and in vivo infectivity data [50].

Calculations of ID_50_ and DD_50_ were also carried out using the R scripts published by Weir 2017 [48] for the exponential and approximate beta–Poisson models (R Core Team). Briefly, the script uses maximum likelihood estimation to fit the counting data (number of animals with virus shedding/diarrhea) to both theoretical models, either exponential or approximate beta-Poisson due to their biologic plausibility. The process calculates the probability of obtaining the observed data given a theoretical distribution by minimizing the deviance (Y) of each of these fitted models as defined by Equation (5).
Deviance = Y = −2 [lnM2 − lnM1](5)
where lnM1 and lnM2 are the log likelihoods for the full (M2) and restricted (M1) models. Optimized deviance follows a Chi-squared distribution with k-p degrees of freedom, where k is the number of dose levels and p is the number of parameters in the model. The model is rejected if Y > Χ^2^(k-p,α). Credible bands illustrating uncertainty in the dose–response relationship were evaluated using Bayesian Markov Chain Monte Carlo (MCMC) in OpenBUGS (Version 3.2.3), as previously carried out elsewhere [51].

## 3. Results

### 3.1. Assessment of Infection Status in Gn Pigs

Infection caused by HuNoV Cin-2 was defined by the presence of viral RNA quantified using RT-qPCR carried out on samples isolated from rectal swabs collected from PID 1 to 7. The percentage of affected animals, mean days to onset, duration, peak titer and area under the curve (AUC) of virus shed in feces of pigs in each of the seven dose groups are summarized in Table 1. An increase in inoculation dose was positively associated with a shorter incubation period (Figure 1A), which coincided with the observed increased duration (Figure 1B) as well as increased overall virus shedding in feces measured by AUC (Figure 1C). All pigs belonging to dose groups 1 to 4 shed detectable titers of virus in their feces, while 67% of pigs in dose groups 5 and 6 and 25% of the pigs in dose group 7 shed viruses. It is important to note that pigs in dose group 3 shed a significantly higher (>1 × 10^6^ RNA copies) amount of viruses in their feces (*p* < 0.0001) and shed viruses for a significantly longer duration (>3 days) than pigs in the other groups (*p* < 0.0001) (Figure 1A,B).

### 3.2. Assessment of Diarrhea Status in Gn Pigs

After inoculation, pigs were monitored daily for diarrhea status. The percentage of pigs with diarrhea, the mean duration, the AUC and mean onset day of diarrhea from Cin-2 inoculated pigs are summarized in Table 1. A faster diarrhea onset (Figure 1D), a longer duration of diarrhea (Figure 1E) and a higher cumulative diarrhea score (Figure 1F) were observed in Gn pigs that were inoculated with higher doses of Cin-2. Pigs in dose group 3 that had the highest virus titers shed in their feces also experienced diarrhea for a longer duration of time than the pigs inoculated with other doses. Pigs belonging to dose group 1 experienced diarrhea for the longest duration of time (5 out of 7 days) and had the quickest onset (within 1.5 days after inoculation), indicating severe clinical onset, mimicking that in humans [11,20], while HuNoV-associated gastroenteritis typically lasts for 3–5 days after infection among susceptible populations [52]. Pigs in dose group 6 had a delayed onset to diarrhea, together with lower cumulative fecal consistency scores, demonstrating a milder diarrhea burden. No diarrhea was observed among pigs in dose groups 5 and 7.

### 3.3. Determination of ID_50_ and DD_50_ Using Various Dose–Response Models

The frequencies (%) of pigs shedding viruses and experiencing diarrhea were used to calculate the ID_50_ and DD_50_ doses using different dose–response models. The log_10_ID_50_ and log_10_DD_50_, estimated using the conventional, exponential, and beta-Poisson dose–response calculation methods are presented in Table 2.

### 3.4. Comparison of Infectiousness of HuNoV GII.4/2003 Variants Cin-1 and Cin-2 in Humans and Gn Pigs

We compared the percentage, mean duration and peak day of virus shedding between humans and Gn pigs after inoculation with GII.4/2003 variants of HuNoV (Table 3). In the human adult challenge study [32], 16 out of 23 secretors (70%) showed signs of symptomatic illness along with the presence of HuNoV RNA in stool samples detected by RT-qPCR, when orally inoculated with 5 × 10^4^ RNA copies of the challenge strain. In Gn pigs, a dose of 2 × 10^4^ RNA copies infected 67% of pigs, whereas 8 × 10^4^ and 2 × 10^5^ RNA copies infected 100% of pigs (Table 1). It is observed that pigs inoculated with 2 × 10^5^ RNA copies shed virus for a similar duration of days compared to those of the human volunteers in the study conducted by Frenck and colleagues [32].

### 3.5. Comparison of Methods for Dose–response Analysis of Cin-2

Both exponential and approximate beta-Poisson dose–response models are routinely used for the determination of median infectious and median diarrhea doses for different pathogens due to their biological plausibility. Goodness of fit tests and MCMC demonstrated that both models are effective for the determination of ID_50_ (Figure 2A,B and Table 4), but the approximate beta-Poisson method, based on a lower deviance and Akaike Information Criterion (AIC), was determined to be the best-fitting model for determining the DD_50_ of Cin-2 (Figure 2C,D and Table 4). The ID_50_ and DD_50_ values determined by the approximate beta-Poisson model were 2.57 × 10^3^ RNA copies and 2.09 × 10^4^ RNA copies, respectively.

### 3.6. Determination of an Optimal Challenge Dose

Based on the virus shedding and diarrhea status of infected Gn pigs in this study, we approximated the optimal dose for Cin-2 to be 2 × 10^5^. Our data show that pigs infected at this dose had a mean onset day of 1.3 with viruses shed in feces in large quantities (measured by AUC) for almost the whole duration of the infection period (6.3 days out of 7; Table 1). Moreover, pigs in this dose group also experienced the highest diarrhea burden among all dose groups, with diarrhea starting at 2.8 days after inoculation and occurring for a duration of 4 days. Pigs in this group also had the highest mean cumulative diarrhea score of 9.31 (Table 1).

## 4. Discussion

In this study, we evaluated the ID_50_ and DD_50_ of HuNoV GII.4/2003 Cin-2 variants in Gn pigs, a large animal model that has proven to be capable of replicating HuNoV GII.4-associated disease in humans [17,20]. With the exception of the exponential dose–response model, all data analysis methods yielded ID_50_ estimates between 2400–3400 RNA copies (Table 2). Relative to previous estimates from human trials with Norwalk virus, this is above 1015 RNA copies [25] and comparable to 2800 RNA copies [26]. The exponential dose–response model yielded higher ID_50_ estimates, but also had poorer fit relative to the approximate beta-Poisson model. The DD_50_ values for all methods except the exponential dose–response model (which had poor fit) were within a range of 21,000–38,000 RNA copies. While the data analysis approaches feature varying assumptions, limitations, and degrees of statistical complexity, the somewhat tight clustering of results indicates that the simple classical methods perform adequately for this particular dataset. Further research is required to explore scenarios in which the classical approaches may become unreliable and compare them with more statistically rigorous contemporary approaches.

Since the first visualization of HuNoV by electron microscopy (EM), samples isolated from stool filtrates without extensive processing have always been identified as aggregates [25,53,54], a recent study showed that large fractions of noroviruses isolated from fecal samples resided inside membrane-bound vesicles—such vesicles increase virus stability within the gastrointestinal tract and enhance the virus infectivity [55]. It is critically important to note that all of the analyses performed herein assume that the viruses were disaggregated. The aggregation of viruses in the stock used to prepare administered doses generally leads to a suppressed dose response (e.g., fewer infections at lower doses and a higher ID_50_). Additionally, aggregation compromises the Poisson foundation of the mechanistic exponential and (exact) beta-Poisson dose–response models. Performing these types of human or animal clinical studies with little attention to the aggregation status of the pathogen stock suspension has been the state-of-the-art. As discussed by Teunis and colleagues, the difference in the infectivity of 8ffIIa and 8ffIIb strains of Norwalk virus could be attributed to the aggregation statuses of the viruses [24]. They alluded to this aggregation with the inclusion of an electron micrograph image of a cluster, and attempted to estimate the degree of aggregation from the corresponding dose–response data [25]. However, it has been shown that such inference is not possible due to model non-identifiability limitations associated with the impossibility of concurrently evaluating dose response and estimating an unmeasured aggregation parameter [29,56].

Subsequent to the completion of this dose response experiment in Gn pigs, the virus pool was analyzed to investigate the nature of aggregation of the Cin-2 inoculum. Although EM is not routinely used for laboratory diagnosis of HuNoV infections, it provides a good understanding of whether the virus exists in a state of aggregation. Electron micrograph images (not shown) indicated that the Cin-2 viruses were indeed likely aggregated in both the stool suspension as well as serial dilutions which were prepared in the same fashion as the administered doses. This has several important implications: (1) the calculated ID_50_ and DD_50_ values are specific to the degree and nature of virus aggregation in this experiment, (2) the ID_50_ and DD_50_ for disaggregated viruses are likely to be lower, and (3) exponential and (approximate) beta-Poisson models must be considered as empirical fits for these data without the typically asserted mechanistic meaning of parameters. It is important to note that these limitations apply to any dose–response experiment in which (1) disaggregation of administered pathogens was not confirmed or (2) aggregation was known to exist but not well quantified. For this reason, it has recently been suggested that ethics approval of future dose–response experiments should be contingent upon use of disaggregated pathogens (or pathogens aggregated to a known extent) [56].

The GII.4/2006b variant of HuNoV used in the study by Bui and colleagues was primarily observed to cause illness among the pediatric population in day cares and hospitals [57,58,59]. The median infectious dose of that GII.4/2006b variant was identified to be 6.43 × 10^4^ RNA copies using the RM method, based on the infection status of 33–34 day old Gn pigs [18]. On the other hand, the Farmington Hills virus was responsible for around 64% of cruise ship outbreaks and 45% of land-based outbreaks, most of which occurred in long-term care facilities, schools, and restaurants, in the US in 2002 [60,61,62]. In the current study, using the RM method, we estimated the ID_50_ of Cin-2, a variant of the Farmington Hills virus, to be 2.51 × 10^3^ RNA copies in similar aged pigs (Table 2). This analysis suggests that at least a 25-fold higher virus titer is required to establish infection among 50% of the pigs inoculated with the GII.4/2006b virus as compared to the Cin-2 (Table 5), indicating that Cin-2 is a more infectious strain in Gn pigs based on the available information regarding the two HuNoV variants.

In the human clinical study carried out by Frenck et al., administering a dose of 5 × 10^4^ RNA copies of Cin-1 (the parent GII4/2003 variant of Cin-2) caused infection among 70% of secretor positive individuals [32]. The same challenge virus, when administered at a dose of 4.4 × 10^3^ RNA copies, caused infection among 62.5% and disease (vomiting or diarrhea of mild or greater severity) in 37.5% of the placebo recipients in another phase I vaccine clinical trial by Bernstein et al. [7]. These data from placebo recipients compared to the slightly reduced incidence of infection (54%) and disease (20%) among the vaccine recipients was not statistically significantly different, suggesting that the challenge dose was insufficient to demonstrate the protective effect of the vaccine [7]. In the present study, only 67% of the dose group 5 (2 × 10^4^ RNA copies) pigs shed viruses and 33% had diarrhea, whereas 100% of the pigs in dose groups 1–4 shed viruses and 75% to 100% had diarrhea. These data show rates of infection that are consistent with Frenck and Bernstein [7,32], hence, highlighting the importance of identifying the optimal virus challenge dose for the evaluation of protective efficacy of vaccines in animal models and in humans.

Diarrhea status in this study was determined based on assigning fecal scores with researchers blinded to the dose administered to each group. The conditions established within the Gn lab setting are well controlled [15], preventing the interference of extraneous pathogens or dietary changes that can cause diarrhea or influence the state of infection. The presence of other pathogens/microorganisms in the environment can lead to discrepancies regarding the true cause of diarrhea [63].

There has been continued debate within the research community concerning the reliability of the RM and DB methods and their use [64]. Both methods are heavily reliant on the number of dose groups and their even distribution since they use only the information from two dose groups for the calculation of the median dose, completely overlooking any outliers within the data [40]. Due to the ease of these methods, they are still considered to be valuable tools for a rapid and/or rough calculation of ID_50_ and DD_50_ for non-statisticians [18,37,42,65].

The Spearman–Karber (SK) method enables easy calculation by hand and yields more accurate estimates of the median infectious dose than the RM and DB methods [66,67]. It relies on the symmetric nature of dose–response data to calculate the median infectious dose, and it requires doses with 0% and 100% responses to calculate the median infectious dose. Some dose–response studies do not meet this criterion. Here, all of the dose groups had animals that shed viruses; thus this criterion was not met for ID_50_ calculation. On the other hand, the SK method would be applicable for calculating the DD_50_, since there were dose groups in which all (100%) or none (0%) of the pigs exhibited symptoms.

The widespread availability of computing has enabled the development of dose–response models based on regression or known probabilistic mechanisms which are more accurate and robust than non-parametric methods [44]. Both the exponential and approximate beta-Poisson models are routinely used for the quantitative determination of dose–response relationships [68]. Using logistic regression, the ID_50_ of the original Norwalk virus (GI.1, 8fIIa isolate) challenge pool was predicted to be around 2.8 × 10^3^ RNA copies in secretor-positive individuals [26]. Similarly, in the current study, we estimated the ID_50_ of Cin-2 to be around 5.75 × 10^3^ RNA copies. These data show a similar infection potential between the prototypic GI.1 Norwalk virus and the Cin-2. With the availability of more dose–response studies and the relatability between human studies and Gn pig studies, it could eventually be possible to compare the pathogenesis of different HuNoV genogroups to help further identify similarities and/or differences between different HuNoVs.

Beyond generating an ID_50_, contemporary dose–response models can describe the full range of probability of response (including ID_20_ or ID_80_) and they are accurate at low doses as well [69]. They allow greater flexibility and a wider range of understanding in the estimated probability of infection. Comparing the approximate beta-Poisson and exponential models using the AIC (Table 4) showed the suitability of both these models for calculating the ID_50_, though the approximate beta-Poisson model was identified to be a more suitable model for DD_50_ estimation.

## 5. Conclusions

Gn pigs have a similar susceptibility to HuNoV infections as in humans. These pigs are also capable of exhibiting the full course of HuNoV infection and develop disease as it is presented in humans. Using different dose response calculation methods to analyze the data collected from this study, we provide here an example of how and where classical and contemporary dose–response methods can be best applied in order to accurately determine the ID_50_ and DD_50_ of HuNoVs. Determining the optimal HuNoV dose would help establish consistency in terms of the challenge dose for the evaluation of novel vaccine candidates between preclinical and clinical trials. In conclusion, we have established a reliable model that can be used to test candidate prophylactics and therapeutics prior to clinical trials in humans, and the model is ready to be used for the evaluation of immunogenicity and the protective efficacy of candidate HuNoV vaccines.

## Figures and Tables

**Figure 1 viruses-12-00955-f001:**
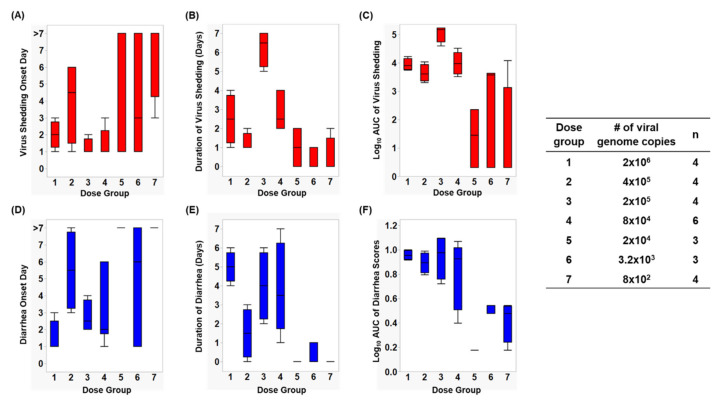
Box and whisker plots showing (**A**) virus shedding onset day, (**B**) duration of virus shedding, (**C**) log_10_ AUC of virus shedding, (**D**) diarrhea onset day, (**E**) duration of diarrhea, and (**F**) log_10_ cumulative diarrhea scores, among each dose group. The maximum and minimum values are denoted by the whisker and the boundaries of each box represent the quartiles with the mean indicated by a black line.

**Figure 2 viruses-12-00955-f002:**
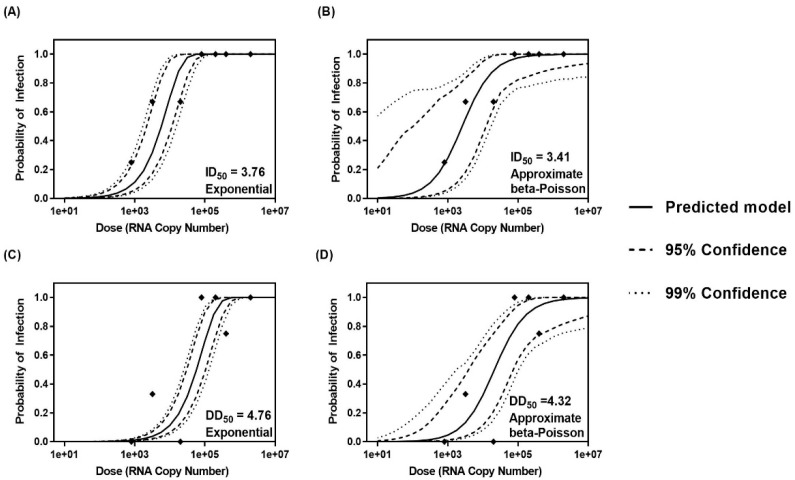
Dose–response curves showing the probability of infection and diarrhea determined by maximum likelihood estimation and credible bands generated by Bayesian Markov Chain Monte Carlo for the exponential (**A**) and (**C**) and approximate beta-Poisson (**B**) and (**D**) models. Dashed lines depict the 95% credible bands, while dotted lines show 99% credible bands. Frequencies of infection and diarrhea determined by infection studies carried out in Gn pigs are depicted as points.

**Table 1 viruses-12-00955-t001:** HuNoV fecal shedding and diarrhea after inoculation of Gn pigs with different doses of Cin-2.

Dose Group	# of Viral Genome Copies	*n*	Virus Shedding				Diarrhea			
			(%) ^a^	Mean Duration Days (SEM) ^c–e^	AUC ^d,f^	Mean Onset Day (SEM) ^c,d^	(%) ^b^	Mean Duration Days (SEM) ^c–e^	AUC ^d,f^	Mean Onset Day (SEM) ^c,d^
**1**	2 × 10^6^	4	4 (100%)	2.5 (0.6) ^BC^	9506 ^B^	2 (0.4) ^B^	4 (100%)	5.0 (0.3) ^A^	9.06 ^A^	1.5 (0.5) ^D^
**2**	4 × 10^5^	4	4 (100%)	1.3 (0.3) ^CD^	5232 ^B^	4 (1.2) ^AB^	3 (75%)	1.3 (0.9) ^ABC^	7.04 ^A^	5.5 (1.2) ^ABC^
**3**	2 × 10^5^	4	4 (100%)	6.3 (0.5) ^A^	126774 ^A^	1.3 (0.3) ^B^	4 (100%)	4.0 (0.9) ^AB^	9.31 ^A^	2.8 (0.5) ^CD^
**4**	8 × 10^4^	6	6 (100%)	2.8 (0.5) ^B^	13495 ^B^	1.5 (0.3) ^B^	6 (100%)	3.8 (1) ^AB^	7.46 ^A^	3.2 (0.9) ^CD^
**5**	2 × 10^4^	3	2 (67%)	1 (0.6) ^B^	93 ^B^	3.3 (2.3) ^AB^	0 (0%)	0.0 (0) ^ABC^	1.50 ^B^	6.3 (1.7) ^AB^
**6**	3.2 × 10^3^	3	2 (67%)	1.3 (0.9) ^BCD^	2667 ^B^	4 (2.1) ^AB^	1 (33%)	1.0 (0) ^BC^	3.17 ^AB^	5 (2.1) ^BC^
**7**	8 × 10^2^	4	1 (25%)	0.5 (0.5) ^D^	2972 ^B^	6.8 (1.3) ^A^	0 (0%)	0.0 (0) ^C^	2.75 ^AB^	8 (0) ^A^

a. Gn pigs were orally inoculated with HuNoV GII.4/2003 variant (Cin-2) at 33–34 days of age. Rectal swabs were collected daily after inoculation from PID 1-7 to determine virus shedding by RT-qPCR. b. Fecal consistency was assessed from PID 1-7 as 0, solid; 1, pasty; 2, semiliquid; 3, liquid. Pigs with daily fecal consistency scores of 2 or greater were considered to be diarrheic. c. SEM, standard error of the mean. d. Numbers in the same column followed by different capital letters (A, B, C, D) differ significantly (Tukey-Kramer HSD, *p* < 0.05); while shared letters indicate no significant difference. e. As pigs were sacrificed on PID 7, some data are right censored (>7). A value of 7 was substituted in mean calculations. f. AUC, area under the curve.

**Table 2 viruses-12-00955-t002:** ID_50_ and DD_50_ calculations of Cin-2.

Method	Log_10_ID_50_	ID_50_	Log_10_DD_50_	DD_50_
**Reed-Muench**				
*Hand calculation* *“skrmdb” R script*	3.40	2.51 × 10^3^	4.58	3.80 × 10^4^
**Dragstedt-Behrens** *“skrmdb” R script*	3.39	2.45 × 10^3^	4.58	3.80 × 10^4^
**Spearman-Karber** *Hand calculation* *“skrmdb” R script* *online calculator*	3.52	3.31 × 10^3^	4.49	3.09 × 10^4^
**Logistic Regression** *online calculator*	3.40	2.51 × 10^3^	4.34	2.18 × 10^4^
**Exponential** ^a^ *R script*	3.76	5.75 × 10^3^	4.76	5.75 × 10^4^
**Approximate Beta-Poisson** ^a^				
*R script*	3.41	2.57 × 10^3^	4.33	2.13 × 10^4^

a. Exponential and Beta-Poisson were determined based on R script used by Weir et al. 2017 [48].

**Table 3 viruses-12-00955-t003:** Comparison of virus shedding in unvaccinated Gn pigs and humans after inoculation with GII.4/2003 HuNoV inoculum.

Host	Age	Challenge Dose	*n*	Virus Shedding (%)	Mean Duration Days (Range) ^c^	Peak Virus Shedding Day (PID)
Human ^a^	19–48 years	5 × 10^4^	23	70	5.2 (2–30)	3
Human ^b^	18–50 years	4.4 × 10^3^	34	76.5	-	-
Human ^b^	18–50 years	4.4 × 10^3^	48	62.5	-	-
Gn pig	33–34 days	2 × 10^4^	3	67	1.0 (1–2)	2
Gn pig	33–34 days	8 × 10^4^	6	100	2.8 (2–4)	3
Gn pig	33–34 days	2 × 10^5^	4	100	6.3 (5–7)	4

a. Data reported by Frenck et al. 2012 [32]. b. Data reported by Bernstein et al. 2015 [7]. c. Virus shedding in human stools was monitored for up to 30 days; and in Gn pigs for up to 7 days post-inoculation.

**Table 4 viruses-12-00955-t004:** Comparison of goodness of fit for the determination of best-fitting ID50 and DD50 model for Cin-2.

	Exponential Model				Approximate Beta Poisson Model				
	r	AIC ^a^	Chi-Squared *p*-Value	Minimized Deviance	α	N50	AIC ^a^	Chi-Squared *p*-Value	Minimized Deviance
**ID50**	1.20 × 10^−4^	10.71	0.71	3.74	0.998	2572	10.68	0.89	1.71
**DD50**	1.20 × 10^−5^	21.46	0.0 ^b^	16.09	0.928	21,340	17.92	0.06	10.5

a. Akaike Information Criterion; b. Rejected due to a poor fit.

**Table 5 viruses-12-00955-t005:** Comparison of Cin-2 to GII.4/2006b challenge pool used in previous Gn pig studies.

Norovirus Variant	Optimal Dose (Viral Genome Copies)	ID_50_ Dose	Method of ID_50_ Determination	
GII.4/2003 Cin-2	2.0 × 10^5^	2.57 × 10^3^	Approximate Beta-Poisson	This study
GII.4/2006b	6.43 × 10^5^	6.43 × 10^4^	Reed-Muench	Bui et al. 2013 [18]; Kocher et al. 2014 [17]; Lei et al. 2016 [16]

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
