# Peer review of "Evaluation of the 50% Infectious Dose of Human Norovirus Cin-2 in Gnotobiotic Pigs: A Comparison of Classical and Contemporary Methods for Endpoint Estimation"

_viruses, 2020, doi:10.3390/v12090955_

Round 1
Reviewer 1 Report
The authors have satisfactorily addressed my comments in their revision, I hence recommend publication in Viruses.
Reviewer 2 Report
Superb work. A clear and concise report. The authors are highly skilled and led by mentors who are among the experts in the field and all are seasoned investigators. They have carefully described their experience with two strains of GII.4 norovirus in gnotobiotic piglets and they report relevant data that undoubtedly will be used by others for building new models and planning advanced experiments. As such, the manuscript is valuable and reflects a tremendous amount of hard work in a field where viral amplification is very difficult to achieve.
This manuscript is a resubmission of an earlier submission. The following is a list of the peer review reports and author responses from that submission.
Round 1
Reviewer 1 Report
Ramesh and colleagues describe efforts to improve a Gn pig infection model for HuNoV by determining condition to acquire ideal ID50 and DD50 of GII.4 those reflect HuNoV infection in human.
They show that severity and mean duration of diarrhea of pigs inoculated with 2 x 105 RNA copies of GII.4 was similar to that of challenged human. The authors show that beta-Poisson model is ideal to eveluate ID50 and DD50 of GII.4. Furthermore, they provide data indicating that aggregations of GII.4 viruses affect infection and shedding of GII.4 in the pigs.
Overall, this is an important study improving methodology of a pig model to understand HuNoV infection.
--
Specific comments
1. Could you provide data of the correlation between an increased dose and a shorter incubation period described in Line 251? It could be calculated from numbers in Table 1, however showing the data would help readers to understand the manuscript easily.
2. This is minor comment. It might be better to unify a term of the dose groups. Two terms exist in the manuscript; "groups X" as in line 254, "Y x 10Z" as in line 255.
3. This relates to the question 1, could you provide graphs of the mean onset day?
4. Take off "supplementary material" in line 283. It is redundant.
5. Based on the whole data in the manuscript, the conclusion that "the infectiousness of HuNoV Cin-1 in Gn pigs and humans are very similar~" in lines 306 seems to be too intense. Comparison of challenge studies between human and pig (Table 3) clearly shows that infectivity of HuNoV in human is higher than that in pigs.
6. The section 3.6 describing the optimal challenge dose of Cin-1 at 2 x 105 seems to be appeared suddenly in the result section. This conclusive table might be moved backward and be referred for the discussion (e.g., to Line 364).
7. It is hard to see aggregations of HuNoV in Figure 4 that supports the conclusion of this study. Could you observe a membrane surrounded aggregations? Have you seen the aggregations in the only indicated groups? The numbers among groups have significance?
8. Discussion of the comparison of statistical analysis to determine the optimal ID50 and DD50 is well written. The reviewer would like to suggest that the authors could add more discussion to compare this pig model and human challenge studies. That might strengthen a significance of this study to develop an animal model for HuNoVs.
Reviewer 2 Report
COMMENTS TO AUTHORS
This manuscript addresses the highly relevant topic of norovirus dose-response. This is evaluated using gnotobiotic pigs and Cin-1 (GII.4/2003) norovirus in an experimental design tailored for the estimation of ID50 (and DD50) and as a step towards standardized testing procedures for vaccine trials. The work bridges the gap between classical approaches for clinicians to analyze such data and the dose-response models that are in common use in quantitative microbial risk assessment by applying several techniques to evaluate ID50 and DD50. The experimental design featured doses in which the viruses were microscopically observed to be aggregated.
***Major General Comments***
The biggest issue with the work is an apparent disconnect between the experimental design and models used that does not become clear until Section 3.7. The analysis is predicated entirely upon models for doses containing disaggregated viruses (or RNA copies) so that the numbers of viruses (or RNA copies) vary among aliquots according to a Poisson distribution. Section 3.7, however, suggests that the viruses (or RNA copies) were aggregated at some point between preparation and administration of the doses. If the viruses (or RNA copies) were not disaggregated during the dose preparation process, then the experimental design and presented data analysis are gravely flawed. Concerns about aggregation and efforts to avoid and quantify it need to be better described in the methods section.
The result of this aggregation problem is three-fold (unless the observed aggregation was re-aggregation after the doses were prepared):
The presented statistical analyses using the disaggregated exponential and (approximate) beta-Poisson models are inappropriately applied, The estimated ID50,DD50 are conditional upon an uncontrolled (or at least weakly characterized) degree of aggregation and have little scientific or predictive value except when viruses happen to be aggregated to the same unknown degree, and The ID50,DD50 for disaggregated viruses (or RNA copies) fundamentally cannot be estimated from these experimental data.Please see Schmidt (2015) and Schmidt et al. (2019) for discussion of structural nonidentifiability caused by an unknown degree of aggregation in norovirus dose-response experiments. Unless the viruses were disaggregated or the degree of aggregation was rigorously quantified such that it may be regarded as a control variable with a precisely known numerical value (unlikely to be possible), ID50 and DD50 will be structurally nonidentifiable in the aggregated exponential dose-response model and practically so in the aggregated exact beta-Poisson model. Even this unlikely scenario would depend upon an assumed log-series distribution of aggregate sizes, which would be rather difficult to validate. If the viruses could not be disaggregated for dose preparation, then a new disaggregated inoculum should have been prepared by repassage. Concerning aggregation in norovirus dose-response experiments, it has very recently been said in Schmidt et al. (2019) that “this is particularly important if experiments require ethics approval: an experimental design that can be foreknown to provide inadequate information about important model parameters should not be approved”.
***Minor General Comments***
It is too ambiguous to refer to the model in equation 4 as the “beta-Poisson” model. While use of this term in the literature usually refers to the approximate form, it is more technically accurate to apply it to the 1F1 form (as done by Messner et al., 2014 [53]). Please use the term “approximate beta-Poisson” model throughout unless the “exact beta-Poisson” (1F1) model is being addressed (e.g. line 195).
Section 2.6.1 is not written clearly enough to follow what exactly is being regressed. Please provide R code in the supplementary content or the line of code defining the glm.
Section 2.6.2 either provides too much detail about methods that are in common use or not enough to facilitate reproduction. There needs to be some mention of use of the skrmdb package in R here.
Section 2.6.2.2 is in particular need of attention to ensure accurate presentation of the dose-response models considered. Details are provided below.
Please remove all material concerning the Akaike Information Criterion. It can be useful for comparing non-nested models, but is inferior to the chi-squared likelihood ratio test for comparing models that are nested as the exponential and (approximate) beta-Poisson models are here.
***Specific Comments***
Line 186 – Is GLM a typo here?
Line 190 - …the probability of survival…
Lines 191-192 – This is worded incorrectly. I believe that you mean “the mechanistic microbial dose response model is equation (3), in which the number of ingested pathogens surviving to initiate infection is binomially distributed with survival probability r”.
Lines 196-202 – Lines 196-197 are worded incorrectly. I believe that you mean “the probability of survival of each pathogen in the host is a random variable with a beta distribution among hosts”. Line 197 is the appropriate place to note that an approximation of this theoretically derived model is used (currently in lines 201-202). Use of an approximation generally requires verification that the approximation is acceptably accurate. Please note the approximation before equation (4) and that the criteria beta>>alpha and beta>>1, where beta=N50/(2^(1/alpha)-1), are satisfied with the model fits obtained from this study (assuming disaggregated doses).
Lines 204-208 – Bullets (a-c) are for the 2F1 aggregated exact beta-Poisson model, not for the 1F1 exact beta-Poisson model or its approximation used here. Bullet (d) applies to both dose-response models considered and can be stated elsewhere (line 188 or line 193).
Lines 210-213 – Line 210-211 is for the 2F1 aggregated exact beta-Poisson model, not for the 1F1 exact beta-Poisson model or its approximation used here. Lines 211-212 are incorrect…there can be variation in the survival probability among pathogens (within a particular host or exposure event) with mean r, and this does not change either of the models used. Line 213 is unnecessarily repetitive.
Line 223 – The model is not fit to the “rate of animals…”, but the “number of animals…”.
Lines 224-227 – Reorganization of this sentence is needed for improved clarity.
Line 228 – These are not “the MLE”, they are “the log-likelihoods”.
Line 229 – k is the number of dose levels/groups.
Line 230 – The restricted model is rejected if…
Line 232 – Please remove the first sentence. The material is repetitive and the definition of p is poorly worded (it is the number of parameters in the model).
Line 235-238 – This method gives a notoriously bad representation of parametric uncertainty (relative to Bayesian MCMC, for example) when there are few data…especially when many of the dose groups have either 0% or 100% infection/illness as is the case here (though the intervals shown in Figure 2 don’t look too bad). The scatterplot of bootstrapping points likely looks very moth-eaten (discretized), and should be shown in supplementary content.
Table 1 – In the n colunn, it should be 4,4,4,6,3,3,4. Also, unless none of the pigs were shedding/diarrheic on Day 7, the durations are right-censored data (because the pigs were euthanized before the infection/diarrhea had come to an end). The mean duration will be biased low if these right-censored data were substituted with a value of 7 days.
Figure 1 – Some of the text is difficult to read, either due to small size, serif font, or distortion from resizing.
Table 2 – The entry for “conventional” RM method needs to be re-checked. I ran the calculations using Tables S1A and S1B (with reference to the 1938 paper [34]) and got answers more consistent with the R package. If two supposedly equivalent procedures yield different results, there needs to be an investigation of the cause (also for the “Beta-Poisson online tool”).
Figure 2 – The text in these figures is much too small and blurry (distortion from resizing).
Line 304 – This finding is quite inconsistent with Teunis et al. (2014), which found shedding often lasts >10 days, even up to almost 80 days. Perhaps this result is specific to Cin-1 and not very representative of HuNoV as a whole?
Lines 311, 404 – These are not logit or logistic regression models. They are not fit by GLM with a logit link function.
Tables 4a and 4b – This may be a suitable place to show the actual MLE parameter values of r, alpha, and N50 (or beta).
Line 336 – Messner et al. (2014) [53] did not demonstrate aggregation of this inoculum (by objective observation in the lab), they assumed it (made it up based on supposition).
Line 411 – Atmar et al. (2014) [29] did not use the 8fIIa isolate as erroneously claimed by Messner et al. (2014) [53] and unfortunately copied in Schmidt (2015)…they used a repassage of the 8fIIa isolate.
Line 444 – 8fIIa/8fIIb
***References***
Schmidt (2015) – https://doi.org/10.1111/risa.12323
Schmidt et al. (2019) – https://doi.org/10.1111/risa.13386
Teunis et al. (2014) - https://doi.org/10.1017/S095026881400274X